# Craftium: Bridging Flexibility and Efficiency for Rich 3D Single- and Multi-Agent Environments

**Mikel Malagón**[1]**, Josu Ceberio** [1]**, Jose A. Lozano**[1,2]

{mikel.malagon,josu.ceberio,ja.lozano}@ehu.eus

[1]**University of the Basque Country UPV/EHU**
[2]**Basque Center for Applied Mathematics (BCAM)**

## Abstract

Advances in large models, reinforcement learning, and open-endedness have accelerated progress toward autonomous agents that can learn and interact in the real world. To achieve this, flexible tools are needed to create rich, yet computationally efficient, environments. While scalable 2D environments fail to address key real-world challenges like 3D navigation and spatial reasoning, more complex 3D environments are computationally expensive and lack features like customizability and multi-agent support. This paper introduces Craftium, a highly customizable and easy-to-use platform for building rich 3D single- and multi-agent environments. We showcase environments of different complexity and nature: from single- and multi-agent tasks to vast worlds with many creatures and biomes, and customizable procedural task generators. Benchmarking shows that Craftium significantly reduces the computational cost of alternatives of similar richness, achieving +2K steps per second more than Minecraft-based frameworks. [1]

## 1 Introduction

Progress in Reinforcement Learning (RL) (Sutton & Barto, 2018), embodied AI (Paolo et al., 2024), and open-ended agents (Hughes et al., 2024) is inherently tied to the environments where agents are trained, evaluated, and analyzed. Each new insight or advancement in the field is supported by an environment that enables its emergence and study. A well-known example is the Arcade Learning Environment (ALE) (Bellemare et al., 2013), which undoubtedly contributed to the advancement of the RL field, marking many of its most important milestones. To name a few: the introduction of the Deep Q-Networks (Mnih et al., 2013), the "infamously difficult Montezuma's Revenge" (Bellemare et al., 2016) that inspired many exploration strategies (Ostrovski et al., 2017; Burda et al., 2019; Badia et al., 2020b), and the first time an agent outperformed humans in all Atari benchmarks (Badia et al., 2020a).

However, as observed throughout the literature, research in these areas is bound to the challenges the employed environments introduce. The researcher often faces a dilemma between computationally efficient but simplistic environments or substantially slower but richer environments. For instance, Continual Reinforcement Learning (CRL) (Abel et al., 2023), Unsupervised Environment Design (UED) (Garcin et al., 2024), and Multi-Agent RL (MARL) (Ying et al., 2023), are greatly affected by the efficiency of the employed environments as they require learning from many tasks or agents. Thus, in these works, experiments are often limited to simple environments as a consequence of the computational cost of employing more complex alternatives (Rigter et al., 2024; Malagon et al., 2024; Rutherford et al., 2024). For example, Craftax relies on 2D grids (Matthews et al., 2024),

---

[1] Code available at https://github.com/mikelma/craftium.

while OMNI-EPIC (Faldor et al., 2025) employs 3D environments of substantially limited diversity compared to alternatives like MineDojo (Fan et al., 2022) or Habitat 3.0 (Puig et al., 2024).

Conversely, works on rich and complex environments (Grbic et al., 2021; Earle et al., 2024; Prasanna et al., 2024; Raad et al., 2024) rely on fully featured video games that have a high computational cost and are closed-source. The best-known of such platforms is Minecraft, which has inspired several single-agent environments and benchmarks over the years (Johnson et al., 2016; Guss et al., 2019; Fan et al., 2022). However, Minecraft is a fully featured and complex 3D game, which makes it substantially more inefficient than simpler alternatives (Wydmuch et al., 2019; Matthews et al., 2024). Furthermore, its closed-source nature greatly limits its flexibility, hindering its application to problems beyond "classic" RL, like CRL, MARL, and UED.

Another important issue that especially affects research in these areas is the lack of flexibility in the environments. Commonly used environments offer no customization or limited possibilities, often restricted to a set of predefined parameters, such as difficulty level or the number of enemies. Among others, these environments include: ALE (Machado et al., 2018), MineRL (Guss et al., 2019), ProcGen (Cobbe et al., 2020), MineDojo (Fan et al., 2022), Crafter (Hafner, 2022), and Craftax (Matthews et al., 2024). The lack of flexibility hinders the ability to analyze specific behavior of agents, obstructing algorithmic comparison beyond pure performance benchmarking, which has been shown to be insufficient for RL (Jordan et al., 2024). Although flexible platforms that allow the creation of new and diverse environments exist, these fall into 2D worlds (Bamford et al., 2020; Chevalier-Boisvert et al., 2023; Matthews et al., 2024) or depend on complex Domain Specific Languages (DSL) that make their implementation difficult, while still not being 3D, as is the case with VizDoom (Wydmuch et al., 2019) and MiniHack (Samvelyan et al., 2021).

In this paper, we present Craftium, an easy-to-use platform for creating rich and efficient 3D environments for autonomous agent research. Unlike most complex environment platforms, which are based on video games (e.g., VizDoom is based on ZDoom and MiniHack on NetHack), Craftium is based on a game engine: Luanti (Luanti Team, 2025b). The integration with our modified version of the engine (see Appendix B) allows the easy creation of complex voxel environments[2] using the powerful and greatly documented Lua Modding API (Luanti Team, 2025a) instead of much less popular DSLs, as employed in ZDoom or MiniHack. Lua (Ierusalimschy, 2006) is a Python-like, easy-to-use and understand, mature, and efficient programming language used in many popular tools and projects, e.g., Roblox, World of Warcraft, and Neovim. In Craftium, Lua is used to expose the Luanti engine, allowing vast possibilities for developing custom environments. Moreover, Luanti is open-source and has a vibrant community that has created many games and assets that can be used in Craftium environments (Ward, 2025a), significantly reducing the development cost of complex scenarios. For instance, all the environments shown in Section 3.5 have been implemented in less than 160 lines of code (comments and whitespace included). These environments, later described in Section 3.5, showcase the versatility of the presented framework, from RL and MARL tasks of different nature, customizable procedural environment generators for CRL, UED, and meta-RL (Yu et al., 2020; Rimon et al., 2024) to gigantic procedurally generated open worlds (64K×64K×64K blocks) for research on embodied AI (Paolo et al., 2024) and open-ended agents (Wang et al., 2023). Beyond being flexible, feature-rich, and developer-friendly, we show that Craftium environments run 38× faster than alternatives based on the original Minecraft game, the only platforms that offer similar complexity and richness. Craftium also supports running asynchronous environments in parallel, achieving more than 12K steps per second in this setup. Furthermore, Craftium is the first framework that allows the creation of vast 3D open worlds while supporting multi-agent settings, opening the door to new research lines. Finally, Craftium implements the popular Gymnasium (Towers et al., 2024) and PettingZoo (Terry et al., 2021) interfaces, the modern standard for RL and MARL research respectively, making it compatible with many other libraries and projects (Raffin et al., 2021; Huang et al., 2022b; Serrano-Muñoz et al., 2023).[3] Finally, Craftium is fully open

---

[2]Voxel games use 3D blocks (voxels) to construct and represent the world, allowing players to modify the environment by adding or removing blocks.

[3]These interfaces are general and can be used for learning paradigms beyond RL (e.g., evolutionary algorithms).

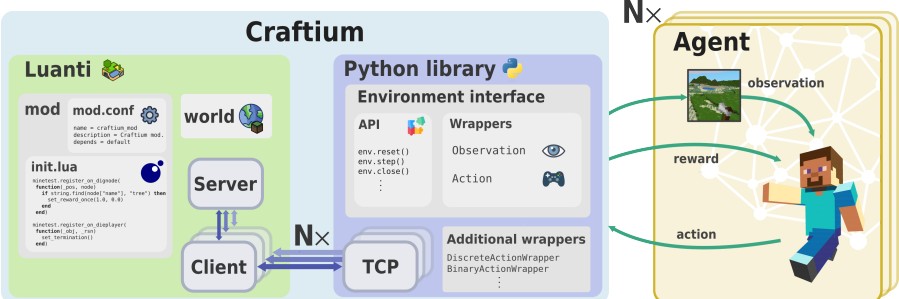

Figure 1: Overview of Craftium's internal architecture. Components denoted with $\times N$ are repeated according to the number of agents (one or more).

source and includes extensive online documentation with many guides, usage examples, tutorials, a detailed reference, and ready-to-use scripts.[1]

## 2 Background: Luanti

Craftium is based on our modified version of the Luanti game engine (refer to Appendix B for details and the list of modifications). Luanti (Luanti Team, 2025b) is a well-known open-source voxel game engine launched for the first time in 2011 that is currently being developed by a vibrant community. Unlike most game engines, Luanti supports modding at its core through its Lua API, allowing fine-grained and real-time access and modification of the internal state of the engine (examples can be found in Section 3.2 and Appendix I). This enables extensive and programmatic customization of its behavior, facilitating the creation, modification, and extension of existing games (environments) using its powerful Modding API (Luanti Team, 2025a; Ward, 2025b). Additionally, Luanti is implemented in C++, a widely adopted programming language known for its high efficiency. Finally, Luanti is supported by an active community that has created hundreds of open, free-to-use games and mods (Ward, 2025a) that are seamlessly loaded in Craftium. For example, community mods are employed in all the environments from Section 3.5.

## 3 Craftium

Craftium follows the architecture illustrated in Figure 1. It consists of two main components: the Luanti engine and the Python environment interface. This interface is the bridge between the environment and the agents. Internally, it handles a communication channel per agent, which connects to Luanti, sending and receiving data such as observations, actions, and rewards. On the other hand, the Luanti server executes the logic of the environment, specified by a file characterizing the 3D world and a script (i.e., mod) that defines its behavior. The Luanti server also synchronizes its clients (one per agent), which handle rendering and communication tasks with the Python library. Finally, note that the original version of Luanti does not support these features, but its open-source nature allowed modifying its source code to support this architecture (see Appendix B).

In the following, Sections 3.1, 3.2, and 3.3 describe Craftium environments, the creation process, and the interface to use them. Respectively, Section 3.4 compares the performance of Craftium with other frameworks. Finally, Section 3.5 showcases the presented framework as a general-purpose environment creation tool across a variety of use cases and fields concerning autonomous agents.

### 3.1 Observations, Actions, and Rewards

**Observations.** In Craftium, observations are images from the agent's point of view. Observations are highly customizable (e.g., size, number of channels, etc.) and can vary between environments. Moreover, Craftium supports many popular techniques, such as frame skipping and frame stacking, that are commonly used throughout the literature (Huang et al., 2022a).

```
1  name = craftium_mod
2  description = My env.
3  depends = default
```

Figure 2: Example configuration file of a mod implementing a Craftium environment.

```
1  core.register_on_dignode(function(ps, block)
2    if string.find(block["name"], "tree") then
3      set_reward_once(1.0, 0.0)
4    end
5  end)
6  core.register_on_dieplayer(function(obj, rn)
7      set_termination()
8  end)
```

Figure 3: Lua script (i.e., mod) implementing basic environment mechanics.

**Actions.** By default, actions are composed of a combination of 21 keyboard actions and a tuple that defines the movement of the mouse, which is mainly used to control the camera. Keyboard-related actions are binary variables with a value of 1 if the key is pressed, and 0 otherwise. The movement of the mouse is defined by the tuple $(\Delta_x, \Delta_y) \in [-1, 1]^2$, where $\Delta_x < 0$ moves the mouse to the left in the horizontal axis and $\Delta_x > 0$ to the right, similarly, $\Delta_y < 0$ moves the mouse downwards in the vertical axis and $\Delta_y > 0$ moves it upwards. Thus, if $\Delta_x = \Delta_y = 0$, the mouse is not moved. See Appendix C.1 for a detailed description of all the possible actions supported in Craftium.

The default action space is designed to be versatile, covering as many use cases as possible: from tasks with complex action sequences (e.g., manual inventory control) to simple navigation environments with a couple of actions (e.g., forward and lateral movement). However, the default action space is overly complex for most tasks: the number of possible keyboard action combinations in the default space is $2^{21}$. Therefore, Craftium allows reducing the action space to the minimal subset required to solve the task at hand, substantially simplifying the learning process of the agent (see Appendix C.2).

**Rewards.** In Craftium, reward functions are defined using Lua scripts (mods are discussed in the next section). Craftium provides a comprehensive set of tools for this purpose, including an extended version of Luanti's Modding API. This functionality is implemented in a modified version of the engine developed specifically for this work, which incorporates additional functions for setting and retrieving reward values and episode termination flags. Some of these functions are shown in the example mod from Section 3.2, while the additional API functions for defining RL environments are detailed in Appendix D. The complete list of modifications to the original Luanti engine can be found in Appendix B.

### 3.2 Creating Custom Environments

Creating a Craftium environment implies two steps: ① generating a *world*: a database with all the information about the virtual environment where the agent will be placed and will interact with; and ② defining the behavior of the environment, such as the reward function and conditions for episode termination. The following lines describe these steps in detail.

① Luanti offers a vast range of possibilities for generating worlds. However, creating a world can be as simple as a few clicks when using one of the many predefined map generators.[4] If finer control over the map generation process is needed, maps can be created using custom scripts. The procedural environment generator presented in Section 3.5.4 is an example of a more complex custom map generation process.

② The next step is to define the behavior of the environment. This is done via mods: user-defined scripts that modify and extend the game engine's behavior, allowing for the creation of custom environments, mechanics, and interactions within the 3D world. A mod has a minimum of two files: a configuration file and a Lua script.

---

[4]Map generators are documented at: https://dev.luanti.org/mapgen.

The **configuration file** contains the mod's metadata. It commonly includes the mod's name, a description, and the list of dependencies (see Figure 2). The **Lua script** is where the environment's mechanics are implemented. Figure 3 illustrates an example script that defines the task of chopping as many trees as possible (presented in Section 3.5.1). Line 1 registers a *callback* function that is called every time the player (i.e., agent) digs a block. In line 2, this function checks if the dug block is part of a tree; if the condition is met, line 3 sets the reward to 1 for that timestep (set_reward_once and other RL related functions are described in Appendix D). Line 7 registers another callback function. In this case, the function is run every time the player dies and calls another function that terminates the episode, in line 8.

Even basic mods, such as the presented example, can be used to generate a wide range of environments. Furthermore, advanced community-made extensions and games can be easily integrated into Craftium, significantly expanding its potential. Section 3.5 highlights some of these possibilities. Refer to Appendix J and to the online documentation[1] for detailed instructions on creating Craftium environments. Finally, note that the creation of Luanti mods is outside the scope of this paper, as comprehensive resources are already available (Luanti Team, 2025a; Ward, 2025b).

## 3.3 Interface

Once created, Craftium environments are used via the Gymnasium (Towers et al., 2024) (single-agent) or PettingZoo (Terry et al., 2021) (multi-agent) interfaces. Both interfaces are open-source and have become the standard interface for RL and MARL environments, providing a unified abstraction over environments that enables interoperability between environments and methods. Just by implementing these interfaces, Craftium is already compatible with many existing tools and projects to train, test, develop, and analyze many algorithms, including but not limited to stable-baselines3 (Raffin et al., 2021), Ray RLlib (Moritz et al., 2018), CleanRL (Huang et al., 2022b), and skrl (Serrano-Muñoz et al., 2023).

```python
import gymnasium as gym
import craftium

env = gym.make("Craftium/Room-v0")
obs, inf = env.reset()
for t in range(5000):
  a = agent(obs)
  obs, r, tm, tc, inf = env.step(a)
  if tm or tc:
    obs, inf = env.reset()
env.close()
```

Figure 4: Python code illustrating the interaction loop between the agent and a Craftium environment using the Gymnasium interface.

Figure 4 illustrates an example using the Gymnasium (single-agent) interface. Note that, PettingZoo employs a very similar interface described in Appendix E. Line 4 loads an example Craftium environment by name (see Section 3.5). Line 6 initiates an episode, obtaining the first observation and a Python dictionary with additional information (e.g., elapsed time). Lines 7-12 implement the agent-environment interaction loop. In line 8, the agent selects an action based on the current observation. The line 9 executes the action specified by the agent, resulting in an observation, a reward, a truncation flag, a termination flag, and a new information dictionary, respectively. The truncation flag indicates if the maximum number of timesteps allowed by the environment is reached, while the termination flag determines if the episode has reached a terminal state (e.g., the player dies). Both flags are checked in line 11, and if one or both of them are true, the episode is restarted in line 12. Finally, the last line closes the environment after the main loop ends.

## 3.4 Performance

As stated in the introduction, computationally efficient environments are key for research on autonomous agents; as such, it has been a focal point of Craftium's development. Figure 5 compares the steps (i.e., interactions) per second obtained by Craftium to VizDoom and MineDojo, well-known environment creation platforms from the litera-

ture. Results show the average of 5 runs in 3 different environments per framework, on a machine with a single NVIDIA A5000 GPU and an Intel Xeon Silver 4309Y CPU. Craftium achieves very competitive results compared to VizDoom, even though VizDoom is based on ZDoom, which is not 3D *per se*, discussed in Appendix G. Comparing Craftium's performance to MineDojo's, we observe that the presented framework achieves +2670 steps per second more. Beyond the single-environment setup, running Craftium environments in parallel notably increases their throughput as shown in Appendix F, reaching over 12K steps per second on the same hardware. By significantly reducing the computational requirements, Craftium enables researchers to conduct large-scale experiments on complex scenarios within their desired domain, supporting advancements in emerging (but especially sensitive to computational cost) areas like CRL, lifelong learning, UED, and open-ended agents. Further details on the benchmark and extended analysis of the results are provided in Appendix F.

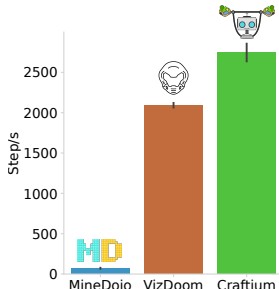

Figure 5: Average steps per second performance comparison.

## 3.5 Illustrative Examples

Craftium is a platform that allows the development of fast and rich 3D environments for all research subfields of autonomous agents, such as RL, MARL, embodied AI, meta-learning, continual RL, and open-endedness. Due to the page limitation, this section highlights Craftium's potential across a few of its vast number of possible use cases: single and multi-agent RL tasks (Sections 3.5.1 and 3.5.2), open-world environments for large multimodal model-based embodied agents (Section 3.5.3), and environment generators for CRL (Section 3.5.4). The aim is to demonstrate the framework's capabilities and provide accessible, well-documented foundations for building custom environments tailored to specific research needs. Note that these examples are merely illustrative and are not to be understood as benchmarks.

### 3.5.1 Example 1: Single-Agent RL

This section provides examples of using Craftium to create single-agent environments for RL. We implement five tasks of diverse nature: simple environments for testing RL algorithms, sparse reward and exploration scenarios, and a challenging survival task. For simplicity, all tasks share the same $64 \times 64$ pixel RGB image observation space. Moreover, the default action space described in Section 3.1 is simplified to only use the necessary actions to solve each task (see Appendix C.2). Figures and extended descriptions of the environments are provided in Appendix H.1.

To complement this example, Figure 6 demonstrates that environments of varying difficulty levels can be designed within Craftium. The figure shows the results of the Proximal Policy Optimization (PPO) algorithm (Schulman et al., 2017) in two of the presented tasks. Results in Figure 6a indicate that the *ChopTree* task can be successfully solved, chopping over 6 trees per episode (+1 for every chopped tree). In *SpidersAttack*, agents are rewarded +1 for every defeated spider, where an additional spider appears in every round (until 5 spiders). As can be seen in Figure 6b, the final episodic return in this task is lower than 1.5, showing that the agents only reach the second of five rounds. See Appendix H.1 for further details and experimental results in the rest of the tasks.

### 3.5.2 Example 2: Multi-Agent RL

This section showcases Craftium's multi-agent capabilities by implementing a MARL environment: a one vs one multi-agent combat environment. Like the tasks from the previous section, this environment employs an RGB image observation space and a simplified discrete action space. Agents are rewarded (+1) when punching other agents and penalized for damage (-0.1). To illustrate an example, we train the agents using self-play (Crandall & Goodrich, 2005), a popular method for this type of competitive scenario (Silver et al., 2017; 2018; Jiang et al., 2024). Results are presented in

Figure 6c, where the policy has been trained to play against itself using PPO. The increasing episodic return curve in the figure shows how the policy learns to fulfill the task. Refer to Appendix H.2 for additional figures and more details on the environment and the learning method.

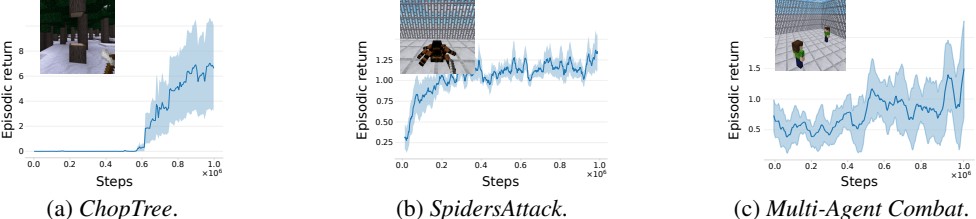

(a) *ChopTree*.  (b) *SpidersAttack*.  (c) *Multi-Agent Combat*.

Figure 6: Episodic return curves obtained by PPO in the single-agent *ChopTree* and *SpidersAttack* tasks, and the multi-agent *MACombat* environment. Results aggregate 5 different runs per task: average is denoted with lines and the standard error with the contour.

### 3.5.3 Example 3: Open-World Environments

This section introduces an open-world environment as an example of a complex scenario for embodied AI. The environment employs the open-source VoxeLibre project (Fleckenstein et al., 2025) for Luanti, which provides a rich and vast environment with many complex interactions, different biomes, animals, plants, or hostile creatures. This section also serves as an example of how community-made games in Luanti can be integrated into Craftium.

Leftmost and center images in Figure 7 illustrate part of the vast and diverse virtual world generated for this environment. Figure 8 presents the skills tree developed for this environment, showing the hierarchical sequence of skills that the agent can develop to reach more complex goals. Every time the agent unlocks a skill of the tool branch (e.g., collect two wood blocks), it receives a reward and new tools (e.g.,

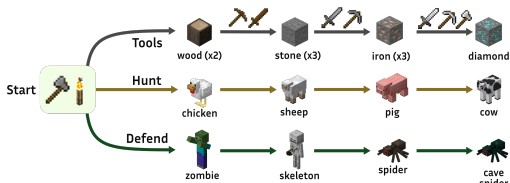

Figure 8: Skills tree of the open-world environment in Section 3.5.3 (see Appendix H.3).

wood pickaxe and sword), while the objective switches to the next skill (e.g., collect two stone blocks). Regarding the hunt and defend branches, the agent receives a reward according to the difficulty of hunting the animal or defeating the monster (refer to Appendix H.3 for details).

To complement this example, the rightmost plot in Figure 7 compares the achievements of PPO+LSTM and an agent based on the open-source large multimodal model LLaVa (Liu et al., 2024a) version 1.6 by Liu et al. (2024b) (with no finetuning to this specific task). Results show that the LLaVa-Agent unlocks the collect wood and stone stages, while PPO+LSTM only completes the first one. Both methods successfully hunt animals and fight some monsters. This example demonstrates Craftium's usage beyond RL, analyzing and evaluating the ability of large multimodal model-based agents to leverage world knowledge to approach complex open-world tasks. Additional information and details are provided in Appendix H.3.

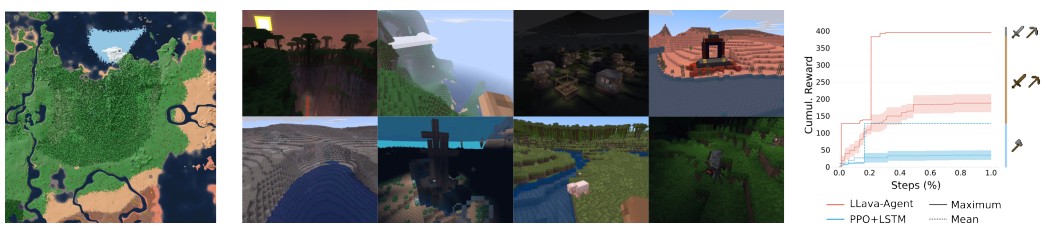

Figure 7: The leftmost picture shows an overview of the map for the open-world environment example. The rightmost plot shows the results of PPO+LSTM and LLava-Agent (zero-shot) in terms of average and best cumulative reward values across 10 repetitions per method (see Appendix H.3).

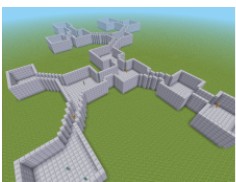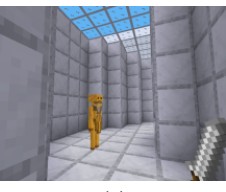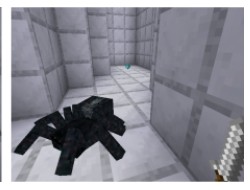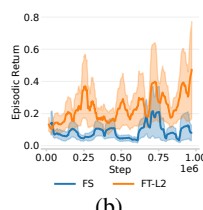

(a)                                                                                    (b)

Figure 9: Images in (a) show examples of the procedurally generated environments from Section 3.5.4. From left to right: the dungeon's top view, and two observations from the agent's perspective. The plot in (b) shows the episodic return curves of FS and FT-L2 in the fourth CRL task.

### 3.5.4 Example 4: Procedural Environment Generation for CRL

This section demonstrates Craftium's versatility by implementing a procedural environment generator that automatically constructs a sequence of increasingly difficult tasks. While such a generator has broad applications, including meta-RL (Dennis et al., 2020), open-endedness (Wang et al., 2023), and UED (Rigter et al., 2024), we focus on a use case in CRL (Abel et al., 2023). In CRL, agents interact with a sequence of environments, each constrained by a timestep budget, and are expected to leverage prior experience to solve new tasks efficiently. Existing approaches typically rely on manually designed task sequences, which limits their scalability and diversity, and rely on repetitive patterns to extend them (e.g., Wołczyk et al. (2021) and Tomilin et al. (2023)). In contrast, our generator enables the automatic creation of diverse task sequences with controlled difficulty, showing how Craftium could be used to overcome these limitations. Conditioned on some input parameters, the generator procedurally constructs labyrinthic 3D dungeons populated with hostile enemies. The agent has to navigate these dungeons, survive, and reach its objective, a diamond. Rewards are assigned as +10 for collecting the diamond, +0.5 for defeating an enemy, and 0 otherwise. In this example, we generate 10 environments of increasing difficulty. Figure 9a illustrates a generated environment, while Appendix H.4 provides further details on the generator.

To complement this example, we train two agents: one from scratch on each task in the sequence (FS) and another that continuously fine-tunes the previously learned model using L2 regularization (FT-L2), a common baseline in CRL (Gaya et al., 2023; Wołczyk et al., 2024). As shown in Figure 9b and Appendix H.4 (complete results in the appendix), FT-L2 significantly outperforms the from-scratch baseline in several environments, demonstrating forward knowledge transfer across the generated tasks.

## 4  Conclusion

This work presents Craftium, an easy-to-use and flexible framework for creating rich and fast 3D environments. Craftium's versatility is showcased in Section 3.5, which shows its application to train and analyze single- and multi-agent RL algorithms, implement open-world environments for complex embodied agent tasks, and procedurally generate environments for CRL. Unlike many alternatives built on top of existing video games, Craftium is based on Luanti, a fully-featured open-source game engine. This analogy is also translated to the presented framework, as it is not a benchmark but a general-purpose tool for creating environments. By leveraging the extensive and well-documented Luanti Modding API (Luanti Team, 2025a), Craftium enables nearly limitless possibilities for the development of custom single- and multi-agent environments. Additionally, Luanti has a vibrant community that has produced numerous games and extensions (Ward, 2025a), which can be easily integrated into Craftium environments. Moreover, its efficient implementation significantly reduces the computational cost of alternatives of comparable richness. As shown in Section 3.4, Craftium achieves over 2K timesteps per second more than MineDojo, and performs competitively with Viz-Doom, even though VizDoom is not fully 3D. Craftium also implements the widely-adopted Gymnasium (Towers et al., 2024) and Petting Zoo (Terry et al., 2021) interfaces, making it compatible with numerous existing tools and projects, such as Huang et al. (2022b) and Raffin et al. (2021). Finally, Craftium is open source and provides extensive documentation, including many practical examples from which users can build environments for their particular research needs.

**Acknowledgments**

We are grateful to Jose A. Pascual for the technical support and to Jon Vadillo and Ainhize Barrainkua for reading preliminary versions of the paper. We also thank the Luanti developers and community for their ongoing efforts to maintain and continuously improve the engine and its ecosystem.

Mikel Malagón acknowledges a predoctoral grant from the Spanish MICIU/AEI with code PREP2022-000309, associated with the research project PID2022-137442NB-I00 funded by the Spanish MICIU/AEI/10.13039/501100011033 and FEDER, EU. Josu Ceberio has been partially supported by the Spanish MICIU/AEI/10.13039/PID2023-149195NB-I00.

This work is also funded through the BCAM Severo Ochoa accreditation CEX2021-001142-S/MICIN/AEI/10.13039/501100011033; and the Research Groups 2022-2025 (IT1504-22), the BERC 2022-2025 program, and Elkartek (KK-2024/00030) from the Basque Government.

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

# Supplementary Materials

*The following content was not necessarily subject to peer review.*

## A  Related Work

Table 1: Popular environment frameworks compared by: number of playable dimensions, procedural generation capabilities, environment creation, whether environments can be programmatically implemented (and not through predefined configuration options), Gymnasium support, multi-agent, and open-world capabilities. We specify the language if a framework allows programmatic implementation of environments, and a red cross otherwise.

| FRAMEWORK | DIMS. | PROC. GEN. | ENV. CREAT. | PROG. DEF. | GYMNASIUM | MARL | OP. WORLD |
|---|---|---|---|---|---|---|---|
| ALE (Bellemare et al., 2013) | 2D | ✗ | ✗ | ✗ | ✔ | ✔ | ✗ |
| DM LAB (Beattie et al., 2016) | **3D** | ✗ | ✔ | **Lua** | ✗ | ✗ | ✗ |
| AI2-THOR (Kolve et al., 2017) | **3D** | ✔ | ✔ | ✗ | ✗ | ✔ | ✗ |
| VIZDOOM (Wydmuch et al., 2019) | 2.5D | ✗ | ✔ | **ZScript** | ✔ | ✔ | ✗ |
| MINERL (Guss et al., 2019) | **3D** | ✔ | ✗ | ✗ | ✗ | ✗ | ✔ |
| NLE (Küttler et al., 2020) | 2D | ✔ | ✗ | ✗ | ✗ | ✗ | ✔ |
| PROCGEN Cobbe et al. (2020) | 2D | ✔ | ✔ | ✗ | ✔ | ✗ | ✗ |
| MINIHACK (Samvelyan et al., 2021) | 2D | ✔ | ✔ | **des-file format** | ✗ | ✗ | ✗ |
| MINEDOJO (Fan et al., 2022) | **3D** | ✔ | ✔ | ✗ | ✗ | ✗ | ✔ |
| HABITAT 3.0 (Puig et al., 2024) | **3D** | ✔ | ✔ | ✗ | ✗ | ✔ | ✗ |
| CRAFTAX (Matthews et al., 2024) | 2D | ✔ | ✗ | ✗ | ✗ | ✗ | ✔ |
| **CRAFTIUM** | **3D** | ✔ | ✔ | **Lua** | ✔ | ✔ | ✔ |

Table 1 includes a comparative overview of popular environment frameworks from the literature.[5] The following lines provide a more extensive discussion of this analysis.

Most of the environments employed in the literature are adaptations of video games that were not originally designed for research (Bellemare et al., 2013; Wydmuch et al., 2019; Guss et al., 2019; Küttler et al., 2020). As a result, they offer limited customization, often restricted to predefined parameters (e.g., number of enemies). Examples include ALE (Machado et al., 2018), MineRL (Guss et al., 2019), and NLE (Küttler et al., 2020). The lack of flexibility hinders their use in various research scenarios, such as designing custom environments to study catastrophic forgetting or analyzing specific behaviors of different learning systems. These limitations have long been recognized, and several frameworks have been proposed that allow the creation of completely new environments. For example, VizDoom (Wydmuch et al., 2019) allows defining environments using ZScript, and MiniHack (Samvelyan et al., 2021) employs the `des-file format` for the same purpose. Both, ZScript and the `des-file format` are *Domain Specific Langauges* (DSL) tailored to the games they originate from (ZDoom and NetHack, respectively). However, DSLs are often purpose-specific and lack the flexibility and functionality of general-purpose programming languages. For instance, the `des-file format` is not a programming language *per se*, just a language to define NetHack levels. Additionally, DSLs often differ significantly from mainstream programming languages, which limits their usability and adoption.

Some frameworks offer customization through the programming languages in which they are implemented, avoiding the limitations of DSLs. For example, Griddly (Bamford et al., 2020) and MiniGrid (Chevalier-Boisvert et al., 2023) offer Python APIs for creating grid-like 2D environments. While grid environments are fast to simulate, they lack the complexity and diversity of more advanced environments like MineRL and VizDoom. Although more complex tasks could be implemented in these frameworks, it would require significant development effort for researchers. Regarding 3D environments, MiniWorld (Chevalier-Boisvert et al., 2023) offers a similar API to MiniGrid but suffers from the same issues regarding the implementation of richer environments.

---

[5]Although the original ProcGen project is unmaintained, the table considers the community rewrite available at `https://github.com/Farama-Foundation/Procgen2`.

On the other hand, the field of embodied AI for robotics has emphasized the importance of visually complex scenarios (Gan et al., 2021), including popular frameworks such as AI2-THOR (Kolve et al., 2017) and Habitat 3.0 (Puig et al., 2024). However, these works focus on accurate physical modeling and photorealism while having limited diversity (mostly including indoor household scenarios) and a lack of open-world environments (Deitke et al., 2022; Gu et al., 2023; Wang et al., 2024). For higher-level cognitive tasks that do not require accurate physics modeling or photorealism, the field has popularly adopted Minecraft—an extremely popular game with rich content and diverse open worlds. Some examples are Malmo (Johnson et al., 2016) and MineRL (Guss et al., 2019), which wrap Minecraft in a Python interface. However, they lack support for task customization or the creation of new environments. More recently, MineDojo (Fan et al., 2022) has greatly improved customization within Minecraft-based environments. Nevertheless, environment creation is constrained by predefined parameters, making scenarios like those in Section 3.5 infeasible to implement (see Appendix I for details) and lacking multi-agent support, which hinders its adoption in this growing field.

## B Modifications to Luanti

Although Luanti is an extremely flexible game engine with extensibility built into its core, we had to modify its source code for this work. As Luanti is a large C++ project with thousands of files, modifications have been thoughtfully introduced to minimize possible conflicts with future updates of the engine. Most of the introduced code is limited to a dedicated `craftium.h` file and some modifications to the `client.cpp` and `game.cpp` files. These are the main modifications that have allowed running autonomous agents in Luanti:

- Implementation of a client that connects to the Python process with the agent's implementation. This is the communication channel from which Luanti sends RGB frames and other timestep data to Python, and Python sends the next actions to be executed.

- Executing the agent's actions as keyboard and mouse commands in Luanti. All actions are translated as virtual keyboard keypresses or mouse movements (for moving the camera and controlling the inventory).

- Extensions to the Luanti API to incorporate vital functionalities for RL environments. Extensions include new Lua functions that implement functions such as setting the episode termination flag or sending reward values.

- Luanti has a client/server architecture, where the server runs the world's logic and the client interfaces with the player (e.g., game control and rendering). However, the asynchronous nature of this architecture introduces issues when using slow agents (e.g., large multi-modal models). For example, the server could update the world many times while the client waits for the agent to return an action. This causes many reproducibility issues and behaviors, such as monsters attacking the player while the client waits for the agent's response. For this purpose, Craftium introduces optional synchronous client/server updates. This ensures that (when needed) the server waits for the client to be updated before continuing with the next update.

- Related to the previous modification, even fast agents (e.g., smaller NNs) can introduce small delays (i.e., lags) to Luanti, for example, when training the model in batches while running the environment. Consequently, we have modified Luanti to avoid being affected by these delays and ensure the environment's and physics's coherence.

- Resetting episodes in complex environments can be time-consuming, sometimes requiring closing and restarting the internal engine of the environment. In consequence, the training time in environments with frequent episode resets (e.g., hard survival games like *SpidersAttack*) increases substantially. To avoid these *hard resets*, we implement several functionalities in Craftium to *soft reset* the environment, not requiring reinitializing the engine. Unlike hard resets, soft resets delegate the reset to the environment itself, to the Lua mod in the case of Craftium (see Appendix D for more details).

## C   Action Space Details

### C.1   Default Action Space

The default action space of Craftium environments is composed of combinations of 21 keyboard actions and mouse movements on the horizontal and vertical axes. Keyboard actions are binary values, where 1 translates to a key press and 0 if not used. Available keyboard commands are listed and described in Table 2. Note that these actions are a subset of the default keyboard controls that Luanti offers[6] and its selection is inspired by the action space of MineRL (Guss et al., 2019). Mouse movements are defined by a tuple (horizontal and vertical movements) of real values in the $[-1, 1]$ interval (see Section 3.1).

Table 2: List of available keyboard actions in Craftium environments, their corresponding key in the default Luanti controls, and their description.

| ACTION | KEY | DESCRIPTION |
| --- | --- | --- |
| Forward | W | Move the player forward. |
| Backward | S | Move the player backward. |
| Left | A | Move the player left. |
| Right | D | Move the player right. |
| Jump | Space | Jump and move up. |
| Aux 1 | E | Run faster. |
| Sneak | Shift | Sneak, move downwards. |
| Zoom | Z | Zoom in at the center of the camera. |
| Dig | Left mouse button | Punch if using a weapon or mine if using a tool. |
| Place | Right mouse button | Use the pointed object if usable, otherwise attempt to build at the pointed block. |
| Drop | Q | Drop the wielded item. |
| Inventory | I | Show/hide inventory. |
| Slot [1-9] | 0-9 | Select the item in the [0-9] position of the hotbar. |

### C.2   Action Wrappers

By default, Craftium environments have a large action space with discrete (binary) and continuous values (see Section 3.1). However, many tasks do not require the complete default action space and can be greatly simplified by considering only the relevant actions to solve the specific task that the environment defines. Consequently, Craftium provides tools for customizing the action space of environments by using Gymnasium Wrappers.[7] Specifically, Craftium implements two wrappers:`BinaryActionWrapper`and `DiscreteActionWrapper`.

`BinaryActionWrapper` allows selecting the subset of keyboard actions (see Table 2 for the complete list) to use in the new action space. This wrapper also simplifies the continuous mouse movement actions by discretizing them into four binary actions: move the mouse left, right, up, and down. The magnitude of these movements can be chosen by the developer. For example, this wrapper allows simplifying the default $\{0, 1\}^{21} \cup [0, 1]^2$ action space into a $\{0, 1\}^3$ space where binary values correspond to: move forward, move mouse right, and move mouse left.

`DiscreteActionWrapper` allows selecting the subset of keyboard actions and discretizes the mouse movement similarly to the previous wrapper. However, in this case, actions are not binary vectors but a single discrete value. Thus, actions can not be combined as in the case of the previ-

---

[6]Some controls like pausing the game or opening the chat have been excluded. For additional information, visit: https://dev.luanti.org/controls/.

[7]Refer to Gymnasium's documentation for more information: https://gymnasium.farama.org/api/wrappers/action_wrappers/.

ous wrapper. Following the previous example, instead of simplifying the default action space into $\{0, 1\}^3$ this wrapper defines the new space as $\{0, 1, 2\}$, where 0 corresponds to move forward, 1 moves the mouse to the right, and 2 moves it to the left.

## D   Extensions to the Luanti Modding API

Luanti counts with an extensive and powerful API (Luanti Team, 2025a) that can be used to modify the behavior of the game engine and create mods or entire games (Ward, 2025a). However, Luanti lacks the functionality to define RL environments by itself. Therefore, Craftium distributes a modified version of the game engine (see Appendix B) that includes additional functionalities in the API to make it possible to implement RL environments from Luanti mods. Table 3 lists and describes the new functions added to the API. Note that besides basic RL environment functionalities, these additions to the API also include functions for soft resetting the environments (see Appendix B for additional information).

Table 3: List of the new functions added to the Luanti API. The "—" character is used to indicate that a function takes no arguments.

| NAME | PARAMETERS | DESCRIPTION |
|---|---|---|
| set_reward | float | Sets the reward value to the given value until another call to a function that modifies the reward is made. |
| get_reward | — | Returns the reward value of the current timestep, and `nil` if not set. |
| set_reward_once | float, float | Sets the reward to the first parameter only for the current timestep, resetting it to the second parameter afterwards. |
| set_termination | — | Sets the termination flag to `true` for the current timestep. |
| get_termination | — | Returns a 1 if the termination flag is set to `true`, 0 otherwise. |
| reset_termination | — | Resets the episode termination falg. |
| get_soft_reset | — | Returns whether the environment should soft reset. |

## E   Using Craftium through the PettingZoo (Multi-Agent) Interface

Figure 10 shows an example use case of the PettingZoo[8] API in Craftium for multi-agent environments. Note that PettingZoo is greatly inspired by Gymnasium and shares many similarities and design choices.[9]

Like the Gymnasium example from Figure 4, the first lines (1-5) instantiate a Craftium environment by name. In this case, *Craftium/MultiAgentCombat-v0* is loaded, corresponding to the multi-agent environment example showcased in Section 3.5.2. Then, line 7 resets the environment to the initial state, initializing Luanti for the first time internally. Next, lines 9-16 define the main agent-environment interaction loop. As defined in line 9, the loop cycles through the agents (two agents for this specific environment). Line 10 obtains the observation, reward, termination/truncation flags, and the information dictionary (similarly to the Gymnasium example). Next, lines 12-13 check if the episode should terminate. If the episode continues, line 15 selects the action for the current agent, and line 16 executes it, running a single environment step for the current agent. Finally, 18 closes the environment, shutting down Luanti and removing any temporary files.

## F   Performance Benchmarks

Due to the page limit constraint of the paper, this section extends Section 3.4 in the main text, including additional details and analyses for different setups.

---

[8]More information at: https://pettingzoo.farama.org/api/aec/.
[9]In fact, both projects are developed under the same Farama foundation, see https://farama.org/.

```python
from craftium import pettingzoo_env

env = pettingzoo_env.env(
    env_name="Craftium/MultiAgentCombat-v0"
)

env.reset()

for agent_id in env.agent_iter():
    observation, reward, termination, truncation, info = env.last()

    if termination or truncation:
        break

    action = agents[agent_id](observation)
    env.step(action)

env.close()
```

Figure 10: Python code illustrating an example multi-agent scenario using the PettingZoo interface in Craftium.

**Single-environment.** To complement the results illustrated in Figure 5, Table 4 provides the exact average and standard deviation values. The measurements aggregate the results of 5 different runs of 1K steps in 3 environments per framework. Note that all environments considered for this experiment were single-agent, as MineDojo does not support multi-agent scenarios[10] and VizDoom does not provide multi-agent environments (although technically supports this setting).[11] The environments were: Speleo, Room, and Spiders Attack for Craftium (see Appendix H.1); *VizdoomHealthGathering-v0*, *VizdoomCorridor-v0*, and *VizdoomDefendCenter-v0* for VizDoom; and *harvest_milk*, *creative:255*, and *Harvest* for MineDojo. In all cases, observations were RGB images, without frameskip, and actions were selected uniformly at random. In the case of MineDojo and Craftium environments observation size was set to $64 \times 64$ pixels, and to $320 \times 240$, as the latter resolution is not available for VizDoom environments.

As can be observed in Table 4, Craftium achieves substantially higher steps per second than the Minecraft alternative, MineDojo. The reasons for such a significant performance gap are many, as both frameworks are complicated systems with many interacting components. One of the most significant differences is the choice of implementation language: MineDojo is based on Minecraft, which is implemented in Java 8, while Craftium relies on Luanti, implemented in C++ and known to perform significantly higher than Java.[12] Another relevant aspect is that Minecraft is a *complete* game, which has grown in complexity over the years, directly affecting the environments implemented on it. As it is a closed-source game, developers are not allowed to modify its source code to remove irrelevant parts of the game for the environment at hand to improve computational efficiency. Contrarily, Luanti is open source and exposes a highly flexible API to modify its behavior. This allows building environments with only the relevant components for the task at hand. Along the same line, the open-source nature of Luanti allowed its modification to tightly integrate it with the proposed framework. For example, to incorporate a system to execute the actions sent from the Python interface as keyboard and mouse commands. Conversely, Minecraft does not allow modifications to its source code, which requires MineRL and MineDojo[13] to include many layers of

---

[10]Revelant discussion at (accessed May 2025): https://github.com/MineDojo/MineDojo/issues/15.

[11]For more details on the multi-agent capabilities of VizDoom (accessed May 2025): https://github.com/Farama-Foundation/ViZDoom/issues/546.

[12]For example, see the performance comparison at https://benchmarksgame-team.pages.debian.net/benchmarksgame/fastest/gpp-java.html.

[13]Note that MineDojo is based on MineRL. Refer to the work by Fan et al. (2022) for details.

complexity to adapt the Minecraft game to the RL setting. Most notably, Minecraft is a game and is not intended to run on a server without a monitor. Therefore, MineRL and MineDojo use an external tool, *Xvfb*.[14] to emulate a monitor without showing any screen output, which causes significant performance drawbacks. This also implies that the X11 windowing system[15] is installed, which is not often the case in HPC clusters.

Table 4: Average and standard deviation values obtained in the environment framework performance comparison conducted in Section 3.4.

| FRAMEWORK | STEP/S |
|---|---|
| CRAFTIUM | **2746.69±230.41** |
| VIZDOOM | 2091.91±59.03 |
| MINEDOJO | 71.87±11.82 |

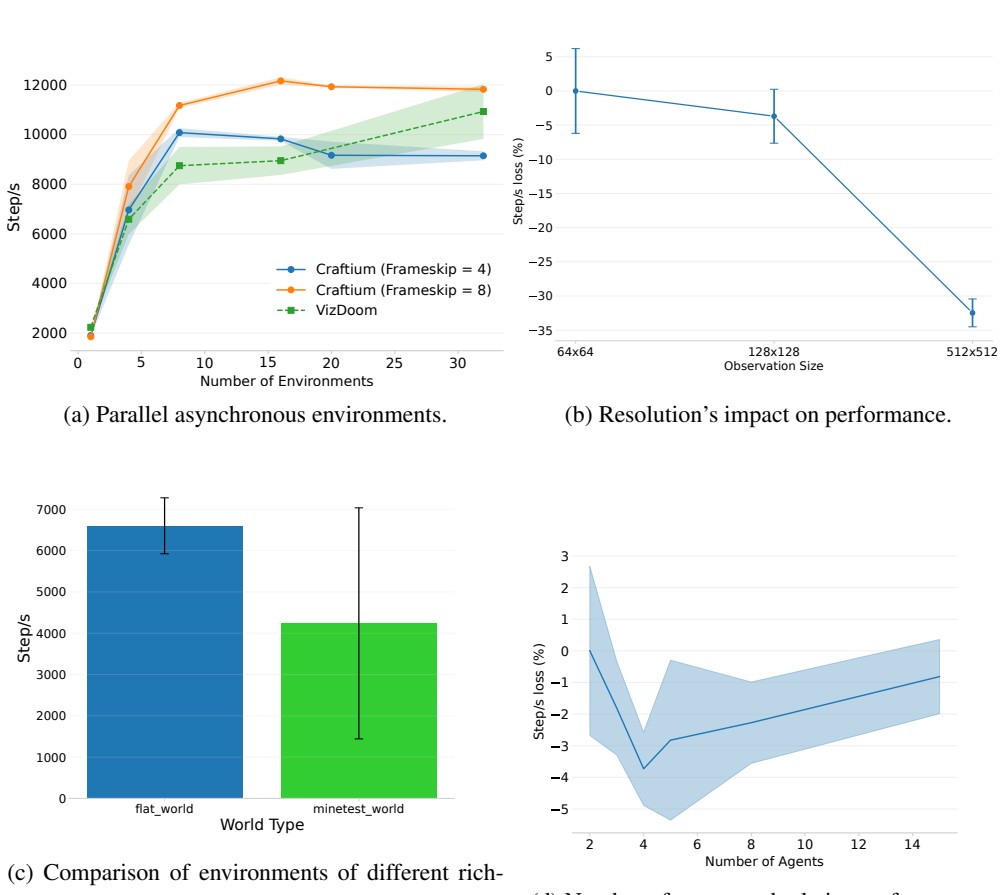

(a) Parallel asynchronous environments.

(b) Resolution's impact on performance.

(c) Comparison of environments of different richness.

(d) Number of agents and relative performance.

Figure 11: Additional performance benchmarks and comparisons. All figures above aggregate the results from 5 repetitions per setup, running 1K time steps on each.

**Parallel environments.** Beyond single environment setups, Figure 11a compares Craftium and VizDoom in vectorized (asynchronous) environments, popularly employed in many on-policy RL methods (e.g., A2C or PPO). As can be observed, Craftium significantly benefits from parallelization, achieving comparable performance to the simpler VizDoom, see Appendix G. In the best setting

---

[14]See https://en.wikipedia.org/wiki/Xvfb.
[15]See https://en.wikipedia.org/wiki/X_Window_System_core_protocol.

for the employed hardware, Craftium surpasses the 12K steps per second using the same number of environments as CPU cores (16 in this case), an unprecedented efficiency for such rich and complex 3D environments. In contrast, MineDojo lacks parallel environment support, and despite efforts, we could not include this framework in Figure 11a.[16] This issue makes MineDojo impractical for many research scenarios, where learning from parallel environments can significantly enhance performance and reduce costs.

**Observation size.** Although many RL tasks might require relatively small observation resolution, as the $64 \times 64$ pixel resolution employed in Section 3.5, some applications might require larger observation sizes, such as large multimodal model-based agents (see Section 3.5.3). Figure 11b shows Craftium's loss in performance for various observation sizes relative to the steps per second achieved with the $64 \times 64$ pixel resolution. As can be seen, Craftium's step per second loss for $128 \times 128$ pixel observations is minimal: less than 5% compared to the baseline performance with $64 \times 64$ pixels. For larger resolutions, $512 \times 512$ pixels in this case, the performance drops considerably, around 33%. However, in such cases, the performance bottleneck is likely in the model processing the images (e.g., a VLM) rather than in Craftium itself.

**Environment's complexity.** Another important aspect that impacts an environment's performance is its complexity or richness. Note that to ensure a fair analysis and comparison, performance benchmarks in this paper consider environments of different nature and requirements, see the beginning of Appendix F for details. To analyze how richness affects Craftium's performance, Figure 11c benchmarks environments using *worlds* (see Section 3.2) of different complexity. The figure shows the results obtained using two world types: `flat_world`, a simple flat world without procedural generation or biomes,[17] and `minetest_world`, a substantially more complex world with procedural generation, biomes, underground dungeons, plants, etc.[18] Results for `flat_world` were collected using the Room and Spiders Attack environments (see Appendix H.1), while Chop Tree and Speleo were used for `minetest_world`. Finally, four parallel environments were employed for both cases. Observing Figure 11c, we see that the world's complexity affects the Craftium's performance (around 30% in this case). However, Craftium's versatility allows the developer to choose within this richness-performance tradeoff, selecting the relevant parts for their specific needs, while discarding unnecessary complexities: enabling procedural generation or not, including animals or NPCs, additional biomes, etc.

**Number of agents.** Regarding Craftium's multi-agent capabilities, in Figure 11d we analyze how the number of agents operating in the same environment impacts performance. The figure analyzes the loss in steps per second as the number of agents increases; for each agent, not in total,[19] and relative to the steps per second achieved with two agents. As can be seen, although the number of agents might decrease the relative steps per second performance, the maximum average loss is lower than 4%. Moreover, the figure shows no noticeable relationship between the two axes: adding more agents has a negligible impact on the relative step per second reached. Finally, at the time of this writing, Craftium supports a maximum number of agents equal to the number of CPU cores of the machine (16 in the case of Figure 11d). This issue limits Craftium's usage on massively multi-agent environments, which we aim to address this issue in future updates.

**Memory usage.** Besides steps per second analyzed in the previous benchmarks, another important efficiency measure is the memory usage of an environment. For instance, memory requirements directly limit the number of parallel environments that can be employed (as studied in the paragraph above on parallel environments). To fairly compare Craftium to VizDoom and MineDojo, we analyzed the memory usage of these frameworks across different tasks, the same ones as in

---

[16]See https://github.com/MineDojo/MineDojo/issues/96 (accessed May 2025).

[17]See https://content.luanti.org/packages/srifqi/superflat/.

[18]More information at https://content.luanti.org/packages/Luanti/minetest_game/.

[19]The relative performance is computed as total steps per second (running the agents in serial, not in parallel) divided by the number of agents.

the single-environment paragraph at the beginning of this appendix. Results show that Craftium (660MB) is notably lighter than MineDojo (1.7GB), the only framework with comparable environment richness. This result highlights Craftium's capabilities to create lightweight environments that avoid extra complexities in tasks that do not require them. Finally, VizDoom (84MB) is the lightest due to its minimalist design. However, reduced memory usage comes at the cost of simplicity, which limits its diversity (e.g., no 3D) and its application to a broad range of research fields (refer to Appendix G for a detailed discussion on the topic).

## G    Limitations of 2.5D Environments

VizDoom is based on ZDoom, a modern and open-source implementation of the original Doom game. The Doom game, released in 1993, employed innovative rendering techniques that made it appear 3D while not having fully three-dimensional scenarios. These rendering techniques, referred to as 2.5D[20] perspective, make VizDoom environments computationally efficient while having some visual features of 3D scenarios. However, 2.5D graphics limits VizDoom environments from an autonomous agent research standpoint compared to fully 3D frameworks such as Craftium. Some of the most notable limitations are the following:

- The agent's viewpoint is restricted to a horizontal plane, preventing it from truly looking up or down.
- Level height (floor and ceiling) is stored in a 2D matrix, making it impossible to create overlapping structures like bridges, floors, or buildings.
- Enemies and objects are 2D sprites that change in size and angle based on the agent's position.

These limitations make VizDoom environments significantly different from more realistic and diverse 3D scenarios as those in Craftium, failing to cover fundamental challenges for autonomous agents that are of interest for current research, e.g., spatial 3D reasoning (Chen et al., 2024) and complex agent-environment interactions (Wang et al., 2023).

Furthermore, 2.5D environments greatly limit the diversity of tasks and scenarios, which is particularly relevant for areas like continual reinforcement learning, unsupervised environment design, and meta-learning, all of which are of growing interest to the research community (Hughes et al., 2024). Craftium is especially relevant to these fields, as it enables diverse, 3D, vast open-world environments that are computationally efficient and support multi-agent scenarios, opening the door to exciting future research directions.

## H    Details on the Illustrative Examples

Due to the size limitations of the main paper, this section includes additional information on the illustrative examples shown in Section 3.5.

### H.1    Environments for Single-Agent RL

All tasks share the same observation space of $64 \times 64$ pixel RGB images. In all cases, the action space has been simplified into a discrete space $a \in \{0, 1, 2, \ldots\}$ as described in Section 3.1 (see Appendix C.1 for details). The simplified action space also introduces a `nop` action (do nothing) to all tasks. The following lines describe the five tasks introduced in this section.

**Chop tree.**    The agent is placed in a dense forest, equipped with a steel axe (see Figure 12a). Every time the agent chops a tree, a positive reward of +1 is given; 0 otherwise. Therefore, the task is to chop as many trees as possible until episode termination. Available actions are nop, move forward, jump, dig (used to chop), and move the mouse left, right, up, and down. Episodes terminate when 2K timesteps are reached.

---

[20]See `https://en.wikipedia.org/wiki/2.5D`.

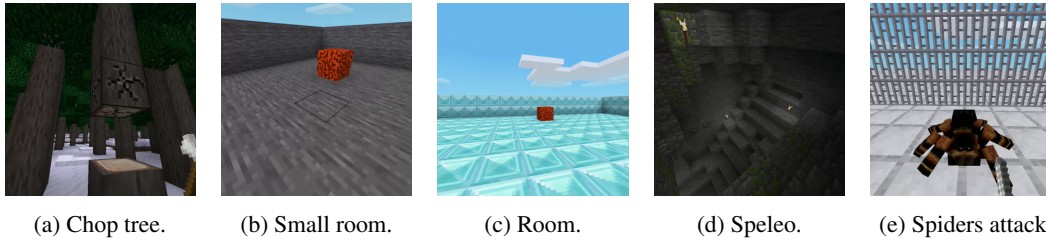

(a) Chop tree.    (b) Small room.    (c) Room.    (d) Speleo.    (e) Spiders attack.

Figure 12: Visualizations of the example environments for single-agent RL in Section 3.5.1.

**Room and small room.**    These tasks present the same objective in different scenarios. In both cases, the agent is placed in one half of a closed room with a red block in the other half of the room. The objective is to reach this block as fast as possible. The difference between both tasks is the size of the room (see Figures 12c and 12b). The reward is constant; all timesteps have a reward value of -1, and the episode terminates when the agent reaches the block. To avoid solving the task by memorization, the initial position of the agent and the red block are randomized in every new episode. Available actions are: move forward, move mouse left, and move mouse right. The timestep budget is 1K in *SmallRoom*, and 2K for the variant with the larger room. Four actions are available: nop, move forward, and move the mouse right and left.

**Speleo.**    The agent is located in a closed cave illuminated with torches (see Figure 12d). The task is to reach the bottom of the cave as fast as possible. For this purpose, the reward at each timestep is the negative altitude (Y-axis position) of the agent. Therefore, the reward increases as the agent goes deeper into the cave. Actions are nop, move forward, jump, and move the mouse left, right, up, and down. Episodes terminate if the agent dies (falling from a great height) or if 3K timesteps are reached.

**Spiders attack.**    The agent is placed in a large cage together with hostile spiders (see Figure 12e), it is equipped with a steel sword, and the objective is to survive. In the beginning, there is a single spider in the cage, but every time all spiders are defeated, a new round starts with one more spider than in the previous one (until 5 spiders). The reward of defeating a siper is +1. Actions are: nop, move forward, move left, move right, jump, attack, and move mouse left, right, up, and down. Finally, episodes terminate if the agent dies or the 4K timestep limit is reached.

Complementing the examples from Section 3.5.1, Figure 13 provides the episodic return curves of PPO in all of the presented tasks. Results aggregate 5 runs per task, where PPO was trained for 1M timesteps each. These experiments are mere examples to complement Section 3.5.1, and thus, no hyperparameter tuning was performed to improve the obtained results. Moreover, the performance in some of the tasks might be substantially improved if more training timesteps are considered.

Regarding the PPO algorithm, we employed the high-quality implementations from CleanRL Huang et al. (2022b). Specifically, the PPO implementation for Atari environments was adapted to Craftium environments, as both observation spaces consist of RGB images and action spaces are discrete (in the case of the environments presented in Section 3.5.1). Moreover, this implementation already considers many details shown to benefit PPO (Huang et al., 2022a). The hyperparameters and CNN network architecture were set according to their default values in the original PPO implementation from CleanRL.[21]

## H.2    Multi-Agent Combat

This section describes the multi-agent environment example from Section 3.5.2 in detail. As can be seen in Figure 14, the scenario consists of a completely flat world, where two agents are

---

[21]Source code of the original PPO implementation: https://github.com/vwxyzjn/cleanrl (commit 8cbca61).

Chop tree Small room Room Speleo Spiders attack

Figure 13: Episodic return curves obtained by PPO in all of the tasks from Section 3.5. Lines aggregate the average values of 5 different seeds per task, while the contour denotes the standard error of the results.

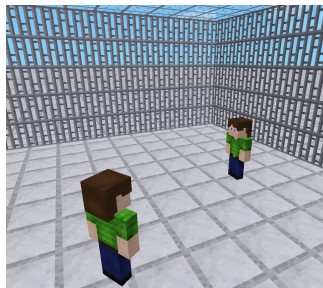

Figure 14: Screenshot of the illustrative multi-agent environment from Section 3.5.2.

placed in a closed jail. Both agents have no items or tools available, and cannot escape the jail. Similarly to the classic single-agent RL task (see Section 3.5.1 and Appendix H.1), observations are $64 \times 64$ RGB images, and the action space consists of a simplified discrete space using the `DiscreteActionWrapper` from Appendix C.2. Specifically, the discrete action space consists of the following actions: nop, forward, left, right, jump, attack, and move the mouse right or left. An agent gets a positive reward (+1) when punching other agents and (-0.1) on damage (i.e., losing one health point). Finally, episodes terminate if the number of health points (initialized to 20) of any of the agents is zero, or the maximum number of timesteps (2K by default) is reached.

Regarding the self-play method employed in Section 3.5.2, we employ the same CNN architecture and PPO algorithm implementation as in the single-agent environment examples from Appendix H.1 (refer to the last part of this appendix for details). In this case, as we employ self-play (Silver et al., 2017), both agents share the same internal NN-based policy, which is updated every 128 steps. Finally, the agents were trained for 1M timesteps using grayscale versions of the observations and frame stacking of 4 frames, resulting in a $4 \times 64 \times 64$ pixel observation space.

### H.3 Open World

In Section 3.5.3 we introduce an open-world environment. In this environment, the agent has to survive and gather resources in an open world based on the open-source VoxeLibre (Fleckenstein et al., 2025) game for Luanti. The environment is designed to have three different tracks: tools, hunt, and defend.

The first, the *Tools* track, consists of 4 different milestones: collect two wood blocks, three stone blocks, three iron blocks, and finally, a diamond block. When the agent unlocks one of the stages (i.e., tasks), it receives a reward and a new set of tools to employ to solve the next task. The reward for completing each of the stages is 128, 256, 1024, and 2048, respectively. Moreover, when the agent unlocks a new stage, it receives a sword and a pickaxe made of the material of the completed stage. For example, if the agent unlocks the *wood* stage (collect two wood blocks), a wood sword and pickaxe are automatically added to its inventory. To simplify solving the first stage of this track, the initial inventory of the agent is composed of a stone axe and 256 torches. The stone axe allows

the agent to more easily chop trees to collect wood, while it also serves to defend from enemies (i.e., monsters) and hunt animals.

Conversely, the *Hunt* and *Defend* tracks are non-sequential. The agent is expected to develop skills to handle increasingly complex scenarios rather than progressing linearly (although this could also be the case). In these tracks, a reward is provided to the agent every time it *punches* an enemy or an animal. In the case of enemies, the reward value is equal to the damage caused by the tool, while in the case of the animals, this value is reduced to half. The motivation behind this particular reward function is the following. If the agent defeats an enemy or hunts an animal, the episodic return obtained by the agent is linear to the life of the enemy or animal. Moreover, the agent is also encouraged to use the correct tool for these tasks. For example, using a sword to fight a monster will provide more reward than using a torch or a pickaxe for the same task.

In Luanti, the *time of day* of the game is linked to the real clock time, where the day/night cycle lasts for 20 minutes by default.[22] In consequence, in this environment, the time of day is set according to the global timestep to maintain consistency and avoid relying on real clock time while training agents. If the latter is not considered, the time of day experienced by the agents could vary depending on the time required by the agent to select an action, which greatly varies depending on its implementation and architecture.

The following lines provide details on the methods used in the experiment from Section 3.5.3. Note that in both cases, the action space of the agents was composed of 18 discrete actions, defined using `DiscreteActionWrapper` from Appendix C.2. The actions are: nop, move forward, backward, left, and right, jump, sneak, dig, place, slot 1, slot 2, slot 3, slot 4, slot 5, move the mouse right, left, up, and down. Slot $[1, \ldots, 5]$ corresponds to the actions of selecting the tool or object in that position of the inventory (i.e., often referred to as the *hotbar*).

**PPO+LSTM.** This method is based on the popular PPO algorithm while employing a convolutional neural network to encode observations and an LSTM module providing memory capabilities to the agent. As the experiments in Appendix H.1, this agent is based on CleanRL's PPO implementations, in this case in PPO+LSTM for Atari games.[23] Similarly, hyperparameters were kept fixed (not optimized), as the purpose of this experiment is to serve as an example. Finally, the observation space for this agent was set to 84×84 of greyscale images using 4 observations for frame stacking.

**LLaVa-Agent.** This agent is based on the open-source large multimodal model (LMM) LLaVa by Liu et al. (2024a), specifically version 1.6 (Liu et al., 2024b). This agent is not intended as a new proposal for LMM for embodied AI, but just as an example of how LMMs can be employed within Craftium environments to solve general tasks by leveraging their world knowledge. For this purpose, LLaVa has been directly employed with no fine-tuning for the open-world environment. Specifically, at each timestep, LLaVa is provided with the current observation (512×512 pixel RGB image) and a short prompt describing the current task. The prompt also includes a list of all available actions, where LLaVa is asked to choose one. Actions are taken by parsing the response from the model, where a random action is chosen if a parsing error occurs, although we observed that this barely happens. The employed prompt has been selected from a set of prompts of different nature listed in Table 6 based on the results from Figure 15, which led to the use of the prompt 1 (ID 1 in Table 6).

Note the `<objective>` placeholder, this is replaced with the text corresponding to the current objective: "is to chop a tree", "is to collect stone", "is to collect iron", or "is to find diamond blocks". This text is automatically placed every time the agent unlocks a stage of the *Tools* branch of the skills tree.

**Details of Figure 7.** The figure aggregates results from 10 different random seeds for each method, PPO+LSTM, and LLaVa-Agent. In the case of LLaVa-Agent, each run was constrained by a 1-hour

---

[22]Additional information at https://wiki.luanti.org/Time_of_day.
[23]The original implementation can be found at: https://github.com/vwxyzjn/cleanrl/ (commit 8cbca61).

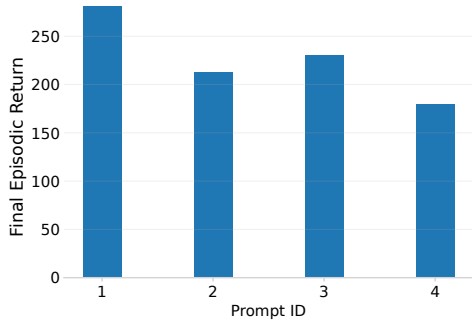

Figure 15: Performance comparison of LLaVa-Agent employing different prompt cadidates from Table 6. The plot aggregates the results from five 1-hour runs for each prompt.

| ID | Prompt |
|---|---|
| 1 | You are a reinforcement learning agent in the Minecraft game. You will be presented with the current observation, and you have to select the next action with the ultimate objective of fulfilling your goal. In this case, the goal `<objective>`. You should fight monsters and hunt animals just as a secondary objective and survival. Available actions are: `<actions>`. From now on, your responses must only contain the name of the action you will take, nothing else. |
| 2 | You are a reinforcement learning agent in the Minecraft game. Your primary objective is: `<objective>`. You must decide the best action based on the current observation. Fighting monsters and hunting animals are secondary tasks and should only be performed when necessary for survival or when they directly contribute to your goal. The available actions are: `<actions>`. Your response must be only the name of the action you will take, with no extra text. |
| 3 | You are an AI reinforcement learning agent in the Minecraft game. Your goal is: `<objective>`. Each step, you receive an observation and must select an action from the following list: `<list-actions>`. Your task is to prioritize the main objective while ensuring survival. Choose the most effective action based on the current observation. You must respond with only the name of the action, nothing else. |
| 4 | You are an autonomous reinforcement learning agent in the Minecraft game. Your mission is to complete the following objective: `<objective>`. Each step, follow a structured decision-making process: (1) Analyze the current observation. (2) Determine whether to focus on the main objective or take necessary survival actions. (3) Choose the best action from: `<actions>`. Your response must be strictly one action name, with no explanations. |

Table 6: Prompt candidates for the LLaVa-based agent in Section 3.5.3: **(1)** direct and imperative tone, **(2)** emphasis on the primary task, **(3)** listing possible actions, and **(4)** structured decision-making.

limit ($\approx$ 7000 prompting iterations per run) and limited to 1M steps in the case of PPO+LSTM. Consequently, the X-axis has been set to the training time percentage to accommodate both cases and for the sake of visualization. Finally, the Y-axis shows the best and average cumulative reward obtained for each method. The latter is made to properly visualize when a method unlocks one of the milestones from the skills tree.

### H.4   Procedural Environment Generation

The procedural environment generation example employs a random dungeon generator implemented for this work. Although the generator can randomly create a vast number of different environments, their reward function is the same. In these environments, the agent is randomly placed (equipped with a sword) in a room and has to navigate a labyrinthic dungeon full of hostile enemies (monsters) to reach the diamond. This process is divided into two steps: ① randomly generate the dungeon's map, represented in ASCII (defined in Appendix H.4.1), and ② build the 3D environment from the map.

① This first step is accomplished by the `RandomMapGen` Python class, which implements the dungeon generation algorithm. Given some input parameters, `RandomMapGen` returns an ASCII representation of the generated map. Internally, `RandomMapGen` first creates the rooms, places the enemies, and locates the objective and the agent's initial position (the agent and the objective are never located in the same room). Then, an iterative algorithm based on repelling forces is used to place the rooms so that none intersect. Secondly, it computes the minimum number of corridors needed to create a map where all rooms are reachable. Finally, it rasterizes the map into its ASCII representation using Bresenham's line algorithm.[24]

The complete list of parameters that `RandomMapGen` accepts is the following:

- Number of rooms of the dungeon.
- Minimum and maximum sizes of the rooms. The final size is randomly selected from this range.
- A dispersion parameter in the $[0, 1]$ range that controls the distance between the rooms.
- Minimum and maximum number of monsters per room. If the minimum is set equal to the maximum, the number of monsters per room is fixed.
- The probability of each monster type being located in one room. `RandomMapGen` considers up to 4 types of different monsters. Monster types are denoted as: `a`, `b`, `c`, or `d`. The specific monster that will be considered for each type is defined by the user in step ②.
- A boolean flag indicating whether monsters can appear in the room selected for the agent's initial position.
- A boolean flag indicating whether to add a ceiling to the map. This option is used when using monsters that can climb over or fly out of the map.

② Once the ASCII map is created, a mod is used to generate the final 3D dungeon inside Luanti. This mod iterates over the characters that compose the map and places the blocks and enemies (referred to as *mobs* in Luanti and gaming terminology, not to be confused with mods) accordingly. The configuration parameters of the mod are the following:

- The ASCII map generated in step ① (or via another process).
- Names of the monsters for types `a`, `b`, `c`, or `d`. Available monsters are described in the documentation of the `mobs_monsters` project.[25]
- The material used for the construction of the dungeons.[26]
- The name of the object to use as the objective (a diamond by default).[27]

---

[24]See https://en.wikipedia.org/wiki/Bresenham%27s_line_algorithm.
[25]Accesible at: https://codeberg.org/tenplus1/mobs_monster.
[26]List of some available materials: https://wiki.luanti.org/Games/Minetest_Game/Nodes.
[27]List of some available items: https://wiki.luanti.org/Games/Minetest_Game/Items.

- The reward of reaching the objective (100 by default).
- The reward of defeating a single monster (1 by default).

### H.4.1 The ASCII Map Format

The ASCII map format has been intentionally designed to be human-readable and to facilitate the implementation of custom procedures to create them (or even be specified by hand). The format consists of 9 possible characters, listed and described in Table 7. As can be seen in Figure 16a, maps are divided into layers, divided by the "−" (dash) character. The first layer is commonly employed to define the floor of the dungeons, while the second defines the walls and the positions of all characters and the objective; the rest of the layers are used for determining the height of the walls.

Table 7: List of characters that comprise the ASCII map format and their meaning.

| Character | Meaning |
| --- | --- |
| (whitespace) | Air block. |
| # | Construction block. Used for the floor and walls. |
| % | Glass block. It can be used for the ceiling. |
| @ | The initial position of the agent. |
| O | Position of the objective. |
| a, b, c, d | Location of a monster of type a, b, c, or d |
| - | New layer. |

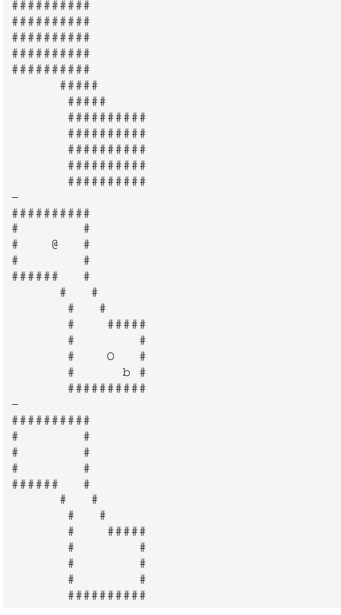

(a) ASCII map representation.

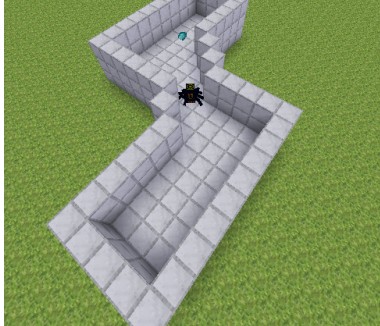

(b) Resulting 3D dungeon environment.

Figure 16: Example ASCII map format of a dungeon environment and the resulting 3D scenario in the Craftium environment. Note the 3D characterizations of the spider (denoted with a in the ASCII map) and the diamond (O in the ASCII map). The ceiling has been removed for the sake of visualization.

### H.5 Environment Sequence for Continual RL

In Section 3.5.4, the procedural environment generator is applied to CRL by defining a sequence of related and increasingly difficult scenarios. Similarly to the examples from Section 3.5.1, the FS (baseline) and FT-L2 methods are based on the PPO implementations from CleanRL. The difference between the FS and FT-L2 is that the latter fine-tunes the model learned in the previous task and uses L2 regularization during training, while the FS always learns a model from scratch. FT-L2 was selected for this example as it has shown significant forward knowledge transfer capabilities in other works (Gaya et al., 2023; Wołczyk et al., 2024; Malagon et al., 2024).

Regarding the observation and action spaces, they have been kept constant across the sequence. The observation space is set to 64×64 pixel greyscale images, with 4 frames for frame stacking, and the same quantity for frame skipping (Huang et al., 2022a). The action space consists of a set of 10 discrete actions: nop, move forward, left, right, jump, attack, move the mouse right, left, and down. Finally, episodes terminate if the health of the agent is exhausted or 5K timesteps are reached.

For the sake of visualization, figure 18 provides a simplified 2D visualization of the environments. Observing the figure, we see that the first two environments employ the same map (with the initial position of the agent and the objective switched). This is intended, as the training time in each environment is low (1M timesteps), thus, the first two environments offer CRL methods a way to learn to reach their objective before more difficult tasks arrive. From the 2nd task onwards, the environments contain two or more monsters, whereas tasks 3 and 4 have a single monster between the agent and the objective (the diamond), and from the 5th task onwards have two or more.

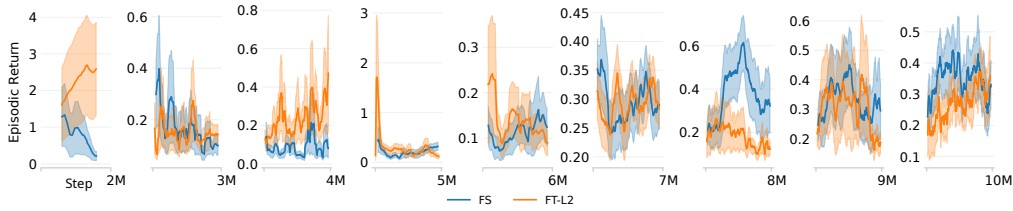

Figure 17: Episodic return curves of the baseline (FS) and FT-L2 over the tasks from the CRL sequence of Section 3.5.4. Note that the first environment is omitted as the FT-L2 is not applied in this case (there is no previous model to fine-tune). See Appendix H.5 for details and Figure 18 for simplified 2D visualizations of all the environments in the sequence.

As can be seen in Figure 17, FT-L2 substantially improves the results of the baseline in the 2nd and 4th, showing considerable forward knowledge transfer between some of the generated environments. Although the final episodic return is lower, environments the 5th and 6th also show some forward knowledge transfer in the first parts of the training.

## I  Environment Creation Flexibility of Craftium and Minecraft-Based Frameworks

Note that the visual similarity between Craftium (and Luanti) and the popular Minecraft game arises from their shared use of voxel-based graphics and sandbox-style[28] environments. However, it is important to clarify that Luanti, as a game engine, is not an implementation or clone of Minecraft, and it serves fundamentally different goals compared to Minecraft, which is a standalone game, see Luanti Wiki (2025).

Besides significant performance improvements, multi-agent support, and a fully open-source nature compared to Minecraft-based alternatives, Craftium also provides an extremely flexible interface for creating new environments via the Luanti Modding API. The flexibility and versatility of this API

---

[28]Sandbox games allow players extensive creative freedom to explore, build, and manipulate the game environment with few constraints or predetermined goals.

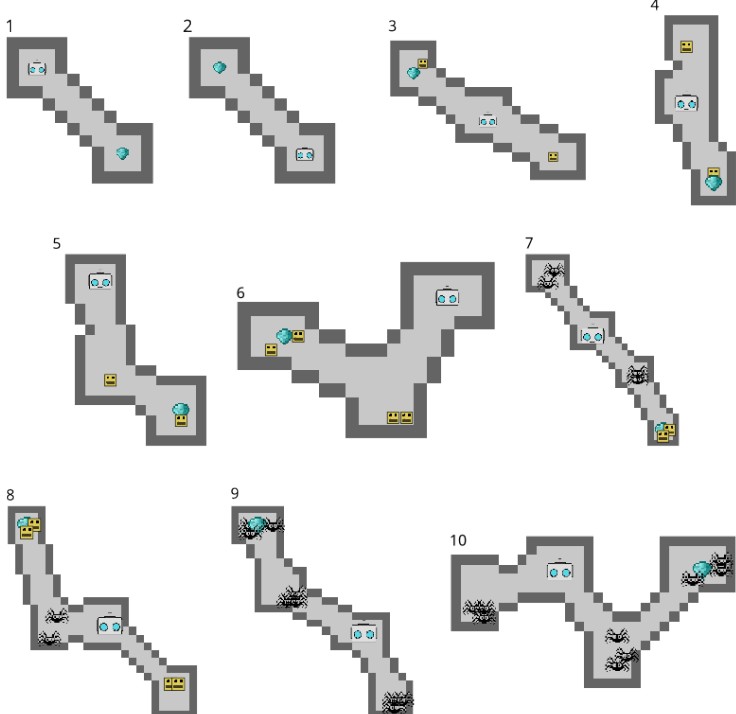

Figure 18: Overview of the maps generated for the CRL environment sequence in Section 3.5.4. Note that these are 2D representations of the environments (for proper visualization) and that the actual environments are 3D, as can be seen in Figure 9a and Figure 16b. The robot indicates the initial position of the agent, while the yellow characters indicate sand monsters, and the black characters denote spiders. Maps have been enumerated with their corresponding position in the CRL sequence.

are demonstrated by the rich and complex environments that can be created with it, see Figure 7, thanks to the wide range of mods created by the community (see Ward (2025a) for examples). This section focuses on showcasing some code examples that directly compare the flexibility of Craftium's API with the MineDojo API to create new environments. Note that we only compare Craftium to MineDojo as it is, currently, the only Minecraft-based framework that allows the creation of custom environments.

One major limitation of MineDojo's API is that although it allows for spawning different Minecraft entities (mobs and items) in a given location, the behavior, aspect, and other properties of the entities are those of Minecraft (the default ones) and cannot be changed. Figure 19 shows how MineDojo allows spawning entities. On the other hand, Craftium leverages the Luanti API, which allows access to the internal state of the game engine, allowing it to change any aspect of it in real time. This is illustrated with an example code in Figure 21 and Figure 22 that show how many properties and behaviors of entities can be modified in Craftium.

Another crucial difference between Craftium's and MineDojo's APIs is the map generation capabilities. MineDojo limits map generation to some predefined scenarios (only 5) and biomes. Figure 20 shows the map customization capabilities of MineDojo. On the other hand, Craftium's API allows the user to define any type of custom biome and combine them in any way.[29] In Figure 23 we show-case a simple example of defining a custom desert biome in Craftium's API. Note that Craftium users can employ any of the vast number of biomes already implemented by the community (some of them illustrated in Figure 7).[30]

---

[29]More information and tutorials at Ward (2025b).

[30]Examples at https://content.luanti.org/packages/?tag=mapgen.

```
1  env.spawn_mobs("spider", [5, 0, 5])
```

Figure 19: **MineDojo.** Although MineDojo allows for spawning entities in some positions, lacks the capability to modify the behavior of entities in any way.

```
1  env = minedojo.make("open-ended", specified_biome="desert")
```

Figure 20: **MineDojo.** MineDojo only allows defining worlds from a set of predefined biomes and scenarios.

```
1  local mob_def = core.registered_entities["mobs_monster:zombie"]
2  mob_def.on_punch = function(self, hitter)
3      hitter:set_hp(hitter:get_hp() + 5)
4  end
```

Figure 21: **Craftium.** Example code demonstrating how the behavior of entities can be modified in Craftium. In this case, the definition of zombies is changed to increase the health of the agent by 5 when successfully attacking a zombie.

```
1  mobs:register_mob("craftium:my_spider", {
2      docile_by_day = false,
3      group_attack = true,
4      type = "monster",
5      passive = false,
6      attack_type = "dogfight",
7      reach = 2,
8      damage = 3,
9      hp_min = 25,
10     hp_max = 25,
11     armor = 200,
12     walk_velocity = 3,
13     run_velocity = 6,
14     jump = false,
15     on_die = function(self, pos)
16         -- Set reward to 1.0 for a single timestep, then reset to 0.0
17         set_reward_once(1.0, 0.0)
18         -- Spawn more spiders
19         num_spiders = num_spiders + 1
20         for i=1,num_spiders do
21             spawn_monster({ x = 3.7 - i, y = 4.5, z = 0.0 })
22         end
23     end
24 })
25
26 local monster = mobs:add_mob(pos, {
27     name = "craftium:my_spider",
28     ignore_count = true,
29 })
```

Figure 22: **Craftium.** Example of a completely custom spider type. Note that we only show a few options of those available: group attack capabilities, health, reach, attack type, armor, velocity, etc. Moreover, a custom behavior is defined to set the reward and spawn more spiders when the spider dies.

```
1  -- Register a custom biome (e.g., desert)
2  core.register_biome({
3      name = "custom_desert",
4      node_top = "default:sand",
5      depth_top = 1,
6      node_filler = "default:stone",
7  })
8
9  -- Generate a random landscape with different biomes
10 core.register_on_generated(function(minp, maxp, blockseed)
11     if math.random() > 0.5 then
12         core.set_biome_area(minp, maxp, "custom_desert")
13     end
14 end)
```

Figure 23: **Craftium.** Example showing how custom biomes can be created and used in Craftium.

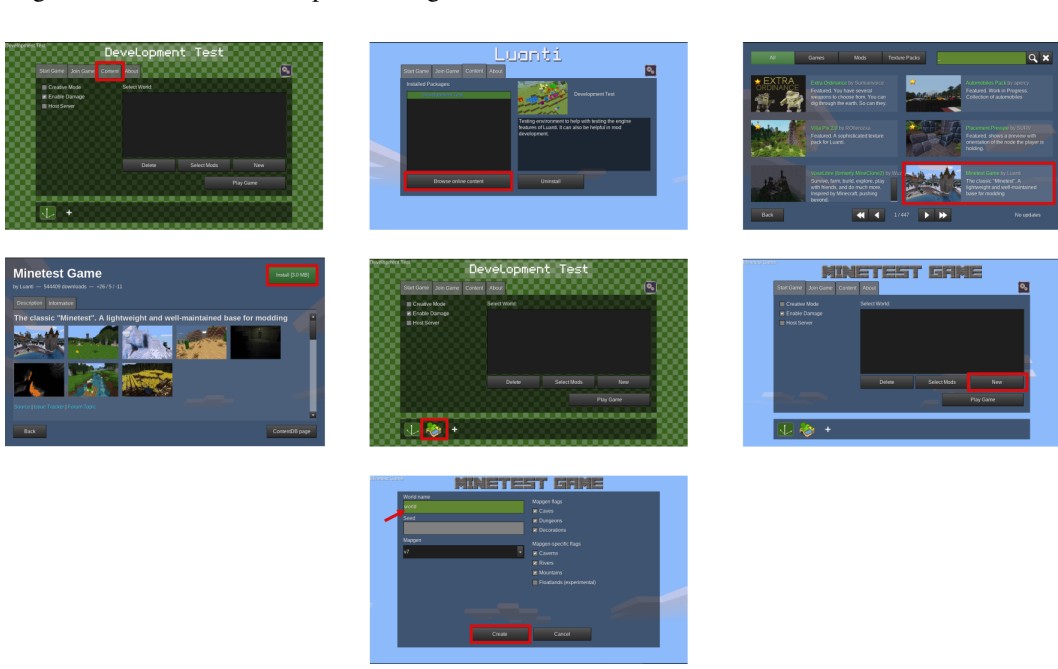

Figure 24: Creating the world using Luanti's graphical menu (left to right, top to bottom). ① Click the *Content* tab in the main menu. ② Click *Browse online content*. ③ Select *Minetest Game*. ④ Click the green *Install* button and wait a few seconds. ⑤ Return to the main menu and click the Minetest logo at the bottom. ⑥ Click *New* to create a world. ⑦ Enter a name (*world* in this tutorial) and click *Create*.

## J  Creating Custom Environments

In the following, we outline the steps to create a custom open-world environment from scratch, where the task is to find the deepest possible cave within a limited number of steps (i.e., episode). Note that the following instructions assume that Craftium is already installed.

① **Creating the world.**   The first step is to create the environment's world. Run the Luanti binary (which should already be built and available in the Craftium installation directory) and follow the instructions in Figure 24. Note that **the first five steps are required only once**; for future environments, follow only steps ⑤ and ⑥ from Figure 24. Close Luanti once the world is created (happens almost instantly).

② **Creating the mod.** Before coding the Craftium environment, set up a directory to store all environment-related data. This directory should contain the game, world, and mod, which we will name `craftium_env`. To create the environment's directory, run the following CLI commands from Craftium's main directory:

```
1 mkdir -r my_env/mods/env_craftium
2 cp -r worlds games my_env
3 echo "load_mod_env_craftium = true" >> my_env/worlds/world/world.mt
```

Figure 25: CLI commands to create and set up the environment's directory.

After running the commands above, we can create the `mod.conf` and `init.lua` files, as described in Section 3.2. In this example, the task is to find the deepest cave possible within an episode. To set up the mod, first, create a file named `mod.conf` inside the `my_env/mods/env_craftium` directory with the contents from Figure 26. Then, create `init.lua` in the same directory using the contents from Figure 27. Although there are many possibilities for the `init.lua` file, in this case, the agent is rewarded based on its negative Y-axis position (depth).

```
1 name = env_craftium
2 description = Craftium environment
3 depends = default
```

Figure 26: Configuration file for the Craftium environment, specifying the mod name and its dependencies (which, in this case, only includes the *default* mod).

```
1  core.register_globalstep(function()
2      -- Get the player's object
3      local player = core.get_connected_players()[1]
4
5      -- Check if the player is connected
6      if player == nil then
7          return
8      end
9      -- Get the player's Y position
10     local y = player:get_pos()[2]
11     -- Set the reward value for the current step
12     set_reward(-y)
13 end)
14
15 -- This function is run every time the player dies
16 core.register_on_dieplayer(function(obj, rn)
17     -- Set the termination flag to true
18     set_termination()
19 end)
```

Figure 27: Example Lua mod. The code defines two callback functions: the first (line 1) runs at every timestep, setting the reward to the player's negative Y-axis position (i.e., depth). The second (line 16) triggers when the player dies (e.g., after a fatal fall) and sets the environment's termination flag to true (see Section 3.3).

③ **Running the environment** With the world and mod set up, the final step is to run the environment. Figure 29 shows a Python script that, when executed in the same directory as the environment (Craftium's main directory in this tutorial), loads the environment and performs random actions

while plotting the current observation at each timestep (see Figure 28). This example simply show-cases the custom environment using a random agent. Note that more complex agents and learning algorithms can be easily integrated by replacing 21 with a call to the desired method.

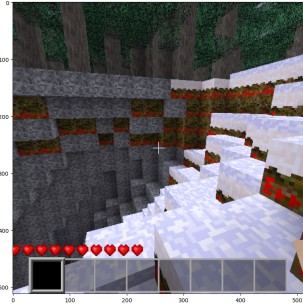

Figure 28: Example output from the script in Figure 29, showing an observation from the environment created in this section.

```python
import matplotlib.pyplot as plt
import craftium
from craftium import CraftiumEnv

env = CraftiumEnv(
    env_dir="my_env",
    obs_width=512,
    obs_height=512,
)

observation, info = env.reset()

ep_ret = 0  # Episodic return
for step in range(100):
    # Display the current observation
    plt.cla()
    plt.imshow(observation)
    plt.pause(0.01)

    # Sample a random action
    action = env.action_space.sample()

    observation, reward, terminated, truncated, _info = env.step(action)

    ep_ret += reward
    print(step, reward, terminated, truncated, ep_ret)

    if terminated or truncated:
        observation, info = env.reset()
        ep_ret = 0

env.close()
```

Figure 29: Example Python script that loads and runs the custom environment with a randomly acting agent.