# OpenReview forum: "Craftium: Bridging Flexibility and Efficiency for Rich 3D Single- and Multi-Agent Environments"
_rl-conference.cc/RLC/2025/Workshop/RLVG — RLVG Workshop - RLC 2025_

### Official Review · Reviewer_qyrQ · 2025-06-15
**Review for Craftium**

**Rating:** 4
**Confidence:** 4

**Summary:**

The authors present Craftium, a highly-configurable simulator capable of producing procedurally generated 3D worlds. Craftium provides egocentric pixels as observations, an action space that emulates mouse-and-keyboard inputs (similar to game playing), and a standard gym-like interface. The appendix contains extensive details on customization options.

**Strengths:**

Craftium's components are explained well and ample details are given. In particular, I am excited about the "large environment" example shown in Example 4 and the Open World details in Appendix H.3.

**Weaknesses:**

I have none to mention.

**Best Paper Nomination:**

Yes

**Claims:**

Yes. The main claims in the paper are about environment-step throughput, and the authors provide hard data that supports these claims.

**Suggestions:**

I do often wonder what sort of limitations there are for these video game style environments. In the introduction, the authors point out that research is influenced (both positively and negatively) by the environments that are used. What might be the consequences if the RL community focuses on video games too much?

---

### Official Review · Reviewer_m1CL · 2025-06-17
**Craftium RL platform**

**Rating:** 4
**Confidence:** 5

**Summary:**

This paper introduces Craftium, an easy-to-use platform for building a highly customizable environment for RL research. This platform allows for creating various environments ranging from simple grid-world-like classic RL environment all the way to complex real-world looking domains. It also integrates existing APIs to allow users to use pre-defined world models, which provides great convenience for researchers. Craftium has the potential to become the new benchmark platform in RL research.

**Strengths:**

A platform that's fully customizable and covers a wide spectrum of environments that satisfy almost any RL research needs.

**Weaknesses:**

Nothing stood out from the paper itself. Perhaps a minor thing is the mixed usage of Lua and Python that might require some initial learning curve for the researcher--I can't judge more until I try it in practice.

**Best Paper Nomination:**

Yes

**Claims:**

The authors provide a detailed infrastructure of Craftium and showcase how it can be used easily. In addition, testing results from classic RL/MARL algorithms indicated the efficiency of the environment. All claims are well supported

**Suggestions:**

This looks like a great platform to use for RL research. It would be cool to benchmark existing RL/MARL algorithms under different environmental settings, from grid-world to open-world.

---

### Decision · Program_Chairs · 2025-06-19

**Decision:**

Accept

**Comment:**

This paper introduces Craftium, a highly customizable and efficient platform for building diverse reinforcement learning environments, ranging from simple grid-worlds to complex, procedurally generated 3D worlds, and integrates existing APIs for convenience.

The reviewers praised the platform for its full customizability and wide spectrum of environment generation capabilities satisfying many RL research needs, and its proven efficiency through benchmarks with classic RL/MARL algorithms. This provides a strong contribution towards the intended goals of the workshop.

A potential weakness is the usage of Lua and Luanti, which might present an initial learning curve for RL researchers. Given the code has not yet been provided, we encourage the authors to provide clear and comprehensive guidance and examples for using the benchmark together with the codebase for presentation at the workshop.